# Black soil layer thickness prediction and soil erosion risk assessment in a small watershed in Northeast China

Keke Xu[1,2,3,4,5], Huimin Dai[2,3,4,5], Xujiao Zhang[1], Chaoqun Chen[2,3,4,5], Kai Liu[2,3,4,5]*, Guanxin Du[6], Cheng Qian[2,3,4,5]

1 School of Earth Sciences and Resources, China University of Geosciences(Beijing), Beijng, China, 2 Shenyang Center of China Geological Survey, Shenyang, Liaoning, China, 3 Key Laboratory of Black Soil Evolution and Ecological Effect, Ministry of Natural Resources, Shenyang, Liaoning, China, 4 Northeast Geological S&T Innovation Center of China Geological Survey, 5 Key Laboratory of Black Soil Evolution and Ecological Effect, Liaoning province, Shenyang, Liaoning, China, 6 School of earth science, Northeast Petroleum University, Daqing, Heilongjiang, China

* 1359422464@qq.com

## Abstract

Black soil has good properties and high fertility. Understanding the spatial distribution of black soil layer thickness is of great significance in promoting regional agricultural development, ecological environmental protection, and soil erosion control. However, traditional soil investigation methods often fail to provide detailed soil thickness information. This study focuses on a small watershed in Northeast China's black soil region. By integrating topographical parameters and vegetation-climate indicators, random forest and kriging methods (classical bayesian, ordinary, and simple) were used to estimate the spatial distribution of thickness of black soil layer. An integrated evaluation framework was developed by combining RUSLE-derived erosion estimates with black soil layer thickness, systematically incorporating both external erosive forces and inherent soil erosion resistance attributes. The results show that the random forest model outperformed the kriging models, with smaller RMSE (34.05 cm) and larger R² (0.57), especially when handling nonlinear, high-dimensional data. The predicted thickness of the black soil layer ranged from 16.2 cm to 107 cm, with a mean of 48.31 cm, closely matching the measured value of 48 cm. Elevation (EL) was found to be the most significant factor affecting the thickness of black soil layer. Soil erosion risk assessment revealed that areas with no risk and low risk accounted for 21.91% and 62.21%, respectively, while medium and high-risk areas made up 15.87% and 0.01%. No-risk areas were soil accumulation zones, while low-risk areas were mainly sloped farmland, where measures like terracing, adjusting crop ridge directions, and planting pedunculated vegetation were recommended. Medium- and high-risk areas should be addressed by returning farmland to forests and implementing engineering practices. This study offers a reference for thickness of black soil layer estimation and provides valuable insights for soil erosion risk management.

**Data availability statement:** All origin data of this paper files are available from the figshare database. (https://doi.org/10.6084/m9.figshare.27964989.v1)

**Funding:** Strategic Priority Research Program of the Chinese Academy of Sciences XDA28020302 to K.L. National Key Research and Development Program of China No.2023YFD1500801 to K.L. Ecological Geological Survey in the Key Areas of Black Soil DD20230089 to K.L. Northeast Geological S&T Innovation Center of China Geological Survey NO.QQJJ2024-27 to K.L.

**Competing interests:** The authors have declared that no competing interests exist.

## Introduction

Northeast China is one of the three major black soil distribution areas in the world, with an area of $1.09 \times 10^6$ km$^2$, of which arable land is $1.85 \times 10^5$ km$^2$. It is an important commercial grain production base, accounting for one-third of the national grain output, and is the "stabilizer" and "ballast stone" of China's grain production [1,2]. Soil thickness affects many important ecological factors such as vegetation growth [3], the hydrologic cycle [4], and surface landscape [5–7]. Understanding soil thickness can also help with ecohydrological research, precision agriculture, and land resource management policies [8]. However, due to long-term overdevelopment and utilization, global climate change, and other reasons, black soil in Northeast China has been degraded to varying degrees. One of the intuitive manifestations is the thinning of the cultivated black soil layer [9]. The black soil layer is the typical marker layer for black soil-type soils and is formed by the mixing of organic matter and minerals accumulated over a long period. It is black or dark black due to the large amount of humus. The texture is dominated by coarse silt and clay. The increase in cultivated land area changes the original land use mode, aggravates the problem of soil erosion, and leads to the decrease of black soil layer thickness and the increase of spatial variability. Soil thickness is the result of the dynamic balance between soil natural development and erosion [1,2]. Black soil in Northeast China has high organic matter and loose structure, while the soil is fragile and vulnerable to erosion. The reduction of natural vegetation further reduces the anti-erosion ability of black soil, with the thickness of black soil decreasing by about 2.22 mm every year [7]. Therefore, it is necessary to estimate the thickness of black soil in northeast China and evaluate the risk of soil erosion.

In a review of large-scale digital mapping of soil properties around the world, soil thickness has poor prediction results [10]. There are three methods to estimate soil thickness: (i) the physical model method, (ii) the interpolation method, and (iii) the environmental correlation method. The physical model method requires relatively dense survey points and the weathering rate of rocks, so it is not suitable for estimating soil thickness in large areas [11]. Interpolation is an objective method for estimating spatial soil thickness, but all parameters related to soil generation and geostatistical models need to be clarified [12,13]. The environmental correlation method uses different statistical models to establish the relationship between soil thickness and environmental variables related to soil occurrence to estimate soil thickness change, which is suitable for both the environment with strong diversity and the environment with strong consistency. Boer et al. [14] reported the use of topographic attributes to map soil depth categories in a large area with high resolution under arid Mediterranean conditions. Kuriakose et al. [11] used empirical and geostatistical methods to predict the spatial thickness distribution of soil, and the results showed that the regression Kriging method had the best effect and could explain 52% of the spatial variability of soil. In southwest China, Yang et al. [15] also used multiple linear regression to predict soil thickness, and the results showed that environmental variables could explain 61.4% of soil spatial variability. However, the above model cannot explain the nonlinear relationship between terrain parameters and soil

thickness [16]. Machine learning models can more intelligently explore nonlinear relationships [17,18]. The random forest model is a machine-learning algorithm based on the decision tree. Combined with the environmental correlation method, it can calculate the interpretation of various environmental variables on soil thickness, avoid overfitting and bias, and require fewer model parameters than other machine learning algorithms [19,20]. Liu et al. [21] used quantile random forest (QRF) combined with detailed characterization of the soil-forming environmental covariates to predict 90 m-resolution soil thickness and other indicators in China. Malone et al. [22] developed a variety of random forest models to predict soil thickness in Australia, and the results were consistent with historical soil survey data across Australia. Vaysse and Lagacherie [23] compared the performance of regression Kriging and QRF in predicting soil thickness in France, and QRF provided a more accurate and interpretable uncertainty prediction model.

The RUSLE (Revised Universal Soil Loss Equation) model has the characteristics of simple structure, easy parameter acquisition, simple calculation, and high precision. It has been widely used in soil erosion modulus estimation at home and abroad [24,25]. The erosion modulus mainly reflects the influence of erosion factors such as precipitation [26], slope [27], and vegetation coverage [28] on soil erosion. These factors drive soil loss by external forcing. Soil thickness, as an intrinsic physical property, plays a pivotal role in determining soil stability, water retention capacity, and erosion resistance [29,30]. Particularly in regions characterized by intense precipitation or steep topographic gradients, thicker soil profiles demonstrate enhanced resistance to erosional processes, effectively mitigating the impact of external erosive forces. Conventional soil erosion assessment predominantly relies on a unidimensional approach utilizing the soil erosion modulus, which solely quantifies erosion intensity per unit time. However, integrating soil thickness parameters establishes a two-dimensional "erosion intensity vs. soil resistance" framework, enabling precise erosion risk classification: (1) high-risk (high erosion modulus, thin soil layers); (2) medium-risk (high modulus, thick layers); (3) low-risk (low modulus, thin layers); and (4) risk-free (low modulus, thick layers).

This study focuses on a small watershed within the typical black soil region of Northeast China. It combines field surveys with model predictions to estimate the thickness of the black soil layer and calculates the importance of various topographical landscape and climate parameters. Based on the prediction results combined with the RUSLE erosion equation, a soil and water erosion risk assessment of the small watershed was conducted.

## Materials and methods

### Study area

The investigated region is situated within the typical black soil zone of northeastern China, representing approximately 12% of global black soil distribution. This ecologically significant area lies in the mid-latitude eastern sector of the Asian continent, characterized by a continental monsoon climate. The climatic regime features mean annual precipitation ranging between 500 mm and 650 mm, with approximately 70%-80% occurring during the growing season (April-September). Mean annual temperatures vary from -5°C to 4°C, exhibiting a latitudinal thermal gradient with decreasing values northward [31]. The distribution areas of the six main soil types in the whole region are dark brown soil, meadow soil, chernozem, black soil, albic soil, and brown soil from large to small [32]. These soils predominantly develop from Quaternary alluvial deposits consisting primarily of sandy gravel and silt-textured materials. The unique combination of loess-derived parent material and continental monsoon climate has facilitated the formation of these characteristic zonal soils through prolonged pedogenetic processes [33,34].

The study area encompasses a 17 km² watershed (46°57'0"N–46°59'0"N, 126°51'07"E–126°56'24"E) located centrally within the typical black soil region (Fig 1). Characterized by a west-to-east topographic gradient, elevations range from 164 m to 256 m. The watershed contains diverse geomorphological features, including low mountains, gullies, and alluvial plains. Land use comprises woodland, dry farmland, and paddy fields. Soils are classified as typical black soils under the Chinese Soil Genetic Classification System, derived from silty Quaternary parent materials. Dominant crops include maize, soybeans, and rice. The area exhibits ecological challenges such as soil erosion and gully development.

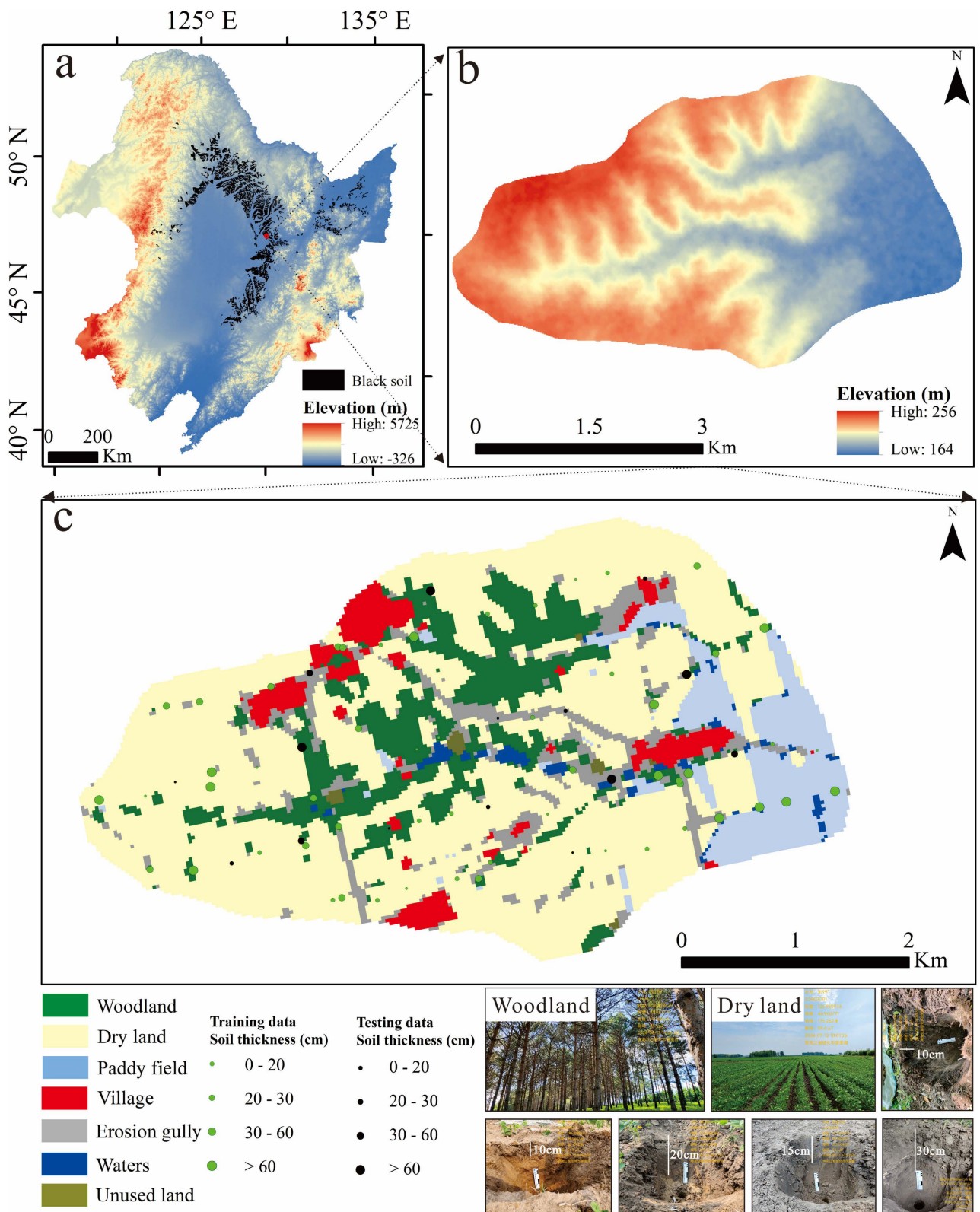

**Fig 1. Location map of the study area. ((a) the black soil region of northeast China and the location of the watershed; (b) digital elevation model of the watershed; (c) land use classification and photographs showing exposed soil sections and land use within the study area).**

## Soil data

The study area was sampled in August 2024. A total of 4 survey routes were set, with 500 meters between each route. The study area exhibits a topographic gradient descending from west to east, with elevated southern and northern regions surrounding a central depression. Therefore, the survey route is in the east-west direction. Sample points were spaced 300 m apart, with additional sample points at sudden land use and terrain changes, resulting in a total of 74 sample points. The study did not involve private land, protected land, endangered or protected species. No specific permissions were required for these locations/activities. Each sampling point location was determined by Global Positioning System technology. The vegetation, land use, landform, and other information around the survey site were recorded during sampling. To minimize the compression of the true thickness of the soil by sampling, samples were collected using 10 cm footfalls at a time, with multiple footfalls until complete profile samples were collected through the black soil layer. Soil profile samples were placed in the field under daylight, and the thickness of black soil layer was determined according to the color, thickness, organic matter content, structure, and salt base saturation of the Chinese soil classification system [35].

## Data sources

The data and sources involved in this paper are shown in Table 1. To facilitate the calculation and analysis of spatial data, the projection coordinate system of data was converted to Krasovsky_1940_Albers, and the raster resolution was unified to 30 m × 30 m.

After radiometric correction and atmospheric correction of Landsat 8 remote sensing images by Envi5.3 software, supervised classification, and visual interpretation were carried out to classify land use types in the study area.

A normalized difference vegetation index (NDVI) was calculated using Landsat OLI/TM remote sensing images in 2024, which reached the maximum vegetation coverage from August to September in the study area. The formula is as follows:

$$NDVI = \frac{\rho_{Nir} - \rho_{Red}}{\rho_{Nir} + \rho_{Red}} \tag{1}$$

$\rho_{Nir}$ represents the reflectance of the near-infrared band in the imagery, while $\rho_{Red}$ represents the reflectance of the red band in the imagery.

The average monthly precipitation is added together to get the average annual precipitation. Dem data is resampled to 30 m resolution. All the above are implemented in the ArcGIS platform. The annual average precipitation data were interpolated to a 30 m resolution using geostatistical methods. Due to the small size of the study area and limited spatial variability in precipitation, errors associated with data resolution conversion were not explicitly considered.

Topographic parameters control the processes from soil to growth and development, as well as the balance of erosion and deposition [36]. Using ArcGIS 10.8 and Sage 7.6.2, seven topographic parameters were obtained for the study area,

**Table 1. Dataset and resources.**

| Name | Type | Spatial Resolution/ Scale | Source |
|------|------|------|------|
| The 2023 Monthly Precipitation Dataset of China | NC | — | National Tibetan Plateau Science Data Center https://data.tpdc.ac.cn |
| The Soil Texture Dataset of China | GRID | 1:100 × 10⁴ | National Tibetan Plateau Science Data Center https://data.tpdc.ac.cn |
| DEM | GRID | 12.5 m | NASA Earth Science Data Website https://nasadaacs.eos.nasa.gov/ |
| Landsat OLI/TM Remote Sensing Imagery | GRID | 30 m | Geospatial Data Cloud https://www.gscloud.cn/ |

including elevation (EL), slope gradient (SG), aspect (A), profile curvature (Cpro), topographic wetness index (TWI), topographic position index (TPI), and horizontal distance to stream (HDTS).

## Methods

In this study, random forest and kriging were used to predict soil thickness and compare the performance of both methods through model validation. In addition, random forests measure the contribution of each environmental variable. Erosion modulus was calculated using the RUSLE equation. The erosion modulus results were combined with the thickness of black soil layer prediction results to jointly assess the risk of soil and water erosion in the black soil layer in the study area.

### Random Forest Model

Random forest is an ensemble learning method that combines multiple decision trees based on decision trees to improve the accuracy, robustness, and generalization ability of the model [20]. Here, we set the number of random forest trees to 50 and 100 to compare the effect, the average tree depth to 10, the number of random sampling variables to 5, and use the out-of-pocket error to evaluate the model performance. Random forest calculation is implemented in ArcgisPro.

### Interpolation model

Kriging interpolation is a statistical interpolation method based on spatial data by establishing a spatial variance function (or covariance function) to describe the spatial correlation between data points and using this information to interpolate in space [37]. Ordinary kriging assumes that the mean of a spatial process is unknown and is an unknown constant in space and estimates the value of each location in the region by using a function of variation (or covariance function), which aims to make the predicted value of the most accurate in the sense of least square error [38]. Simple kriging assumes that the mean of the spatial data is known and constant so that the mean can be used directly in the calculation [39]. Classical Bayesian Kriging combines the Kriging method and Bayesian statistical theory to estimate the parameters of the model through Bayesian inference and describes the uncertainty of the prediction through the way of probability distribution [40].

### Model validation

Out-of-bag (OOB) error is an unbiased estimate that can be used to evaluate the performance of a random forest model because it does not require an additional validation or test set, making it easier to assess the model's predictive power on unknown data [19]. The error evaluation of out-of-bag data has the same accurate effect as K-fold cross-validation [41]. The OOB error and cross-validation results of the random forest model were computationally compared with the cross-validation performance of Kriging interpolation.

The coefficient of determination ($R^2$) and Root Mean Squared Error (RMSE) are used to evaluate the fit of the random forest model predictions and Kriging interpolation predictions.

$$R^2 = 1 - \frac{\sum (y_i - \hat{y}_i)^2}{\sum (y_i - \overline{y}_i)^2}$$

(2)

$y_i$ is the observed value, $\hat{y}_i$ is the predicted value, and $\overline{y}_i$ is the mean of the observed values. The $R^2$ value ranges between 0 and 1.

$$RMSE = \sqrt{\frac{1}{n} \sum_{i=1}^{n} (y_i - \hat{y}_i)^2}$$

(3)

n is the number of samples. The smaller the RMSE value, the smaller the model's prediction error and the better the fit; the larger the value, the greater the prediction error and the worse the fit.

## RUSLE calculation

The Revised Universal Soil Loss Equation (RUSLE) was proposed in 1997 and used to estimate the erosion modulus in the study area [42]:

$$AS = R \times K \times LS \times C \times P \tag{4}$$

In the equation, AS is erosion modulus which represents the annual soil erosion amount per unit area (t/ (hm²·a)); R is the rainfall erosivity factor (MJ·mm/ (hm²·h·a)); K is the soil erodibility factor (t·hm²·h/ (hm²·MJ·mm)); LS is the slope length and steepness factor (dimensionless); C is the vegetation cover factor (dimensionless); and P is the conservation practice factor (dimensionless).

## Results

### Descriptive statistical analysis

A total of 74 survey sites were obtained, and the thickness of black soil ranged from 10 cm to 130 cm, with an average of 48 cm (Table 2). Using 10 cm as the step length to calculate the thickness frequency and frequency of black soil, it can be seen that there is a great difference between samples with different thicknesses (Fig 2). The proportion of 30 cm-40 cm thick black soil is 25.6%. Followed by 40 cm-50 cm, which accounted for 16.02%. The proportion of 20 cm-30 cm and 10 cm-20 cm samples was 13.5% and 12.1%, respectively. The statistical results of the thickness division standard of black soil show that the proportion of "broken yellow" black soil of 0–20 cm is 12.1%, that of 20–30 cm is 25.6%, that of 30–60 cm is 47.3%, and that of more than 60 cm is 27%.

The standard deviation (SD) and coefficient of variation (CV) of the total thickness of black soil layer of the sample points are large, indicating that the spatial variability of the thickness of black soil layer in the basin is large (Table 2). The thickness of black soil in the four terrain groups, namely the top of the slope, the middle of the slope, the bottom of the slope, and the plain, did not pass the variance homogeneity test (Table 3). According to the statistics of the thickness of black soil in different terrains (Table 3), the average thickness of black soil is larger in the slope bottom and plain area, followed by the middle of the slope area, and the thickness of black soil layer at the top of the slope is the thinnest. The large coefficient of variation in the middle slope, bottom slope, and plain area may indicate the complexity of the source of black soil, and the minimum coefficient of variation in the top slope may be related to its denudation environment.

### Correlation analysis

Fig 3 shows the analysis results of the correlation between soil thickness (ST) and environmental variables. TPI, HDTS, NDVI, Cpro, EL, and ST were significantly correlated ($p < 0.01$). The highest correlation between TPI and ST is -0.37. The HDTS correlation coefficient was -0.31. The correlation coefficient between NDVI and ST was 0.28. The negative

**Table 2. Descriptive statistics of thickness of black soil layer of soil samples.**

| Soil samples | Number | Min(cm) | Max(cm) | Mean(cm) | Range(cm) | Percentiles of 90% (cm) | SD(cm) | CV(%) |
|---|---|---|---|---|---|---|---|---|
| Total | 74 | 10 | 130 | 48 | 120 | 110 | 35 | 72 |
| Slope top | 21 | 10 | 40 | 26.2 | 30 | 40 | 9 | 34 |
| Slope middle | 23 | 10 | 130 | 42.2 | 120 | 76 | 29 | 68 |
| Slope bottom | 21 | 10 | 130 | 68.8 | 120 | 110 | 39 | 56 |
| Plains | 9 | 15 | 120 | 65.0 | 105 | 120 | 40 | 61 |

Note: Min, Max, SD, and CV represent the minimum, maximum, standard deviation, and coefficient of variation of thickness of black soil layer.

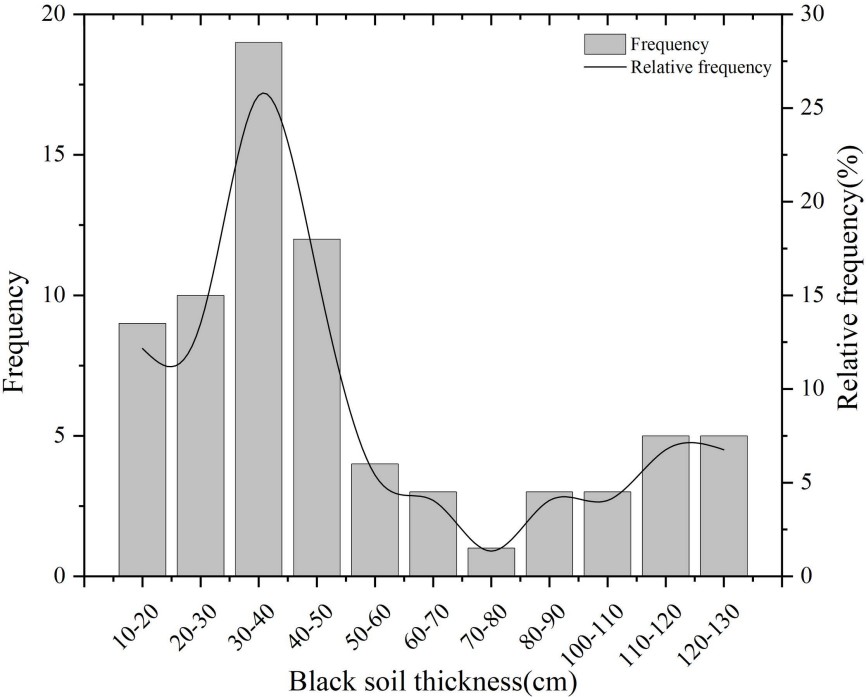

**Fig 2. Histogram of thickness of black soil layer of sample points.**

**Table 3. Analysis of variance of thickness of black soil in different terrain.**

|  | Sum of Squares | Df | Mean Square | F | P value |
|---|---|---|---|---|---|
| Between groups | 22464.165 | 3 | 7488.055 | 8.06 | <0.01 |
| Within groups | 65031.781 | 70 | 929.025 |  |  |
| Total | 87495.946 | 73 |  |  |  |

correlation between EL and ST was -0.10. TWI and ST were positively correlated at a significant level ($p < 0.05$). $P_{ann}$ and A were positively correlated with ST, SG was negatively correlated with ST, and the correlation coefficients were 0.23 and -0.13, respectively.

## Model performance

The number of decision trees in a random forest has a significant impact on model performance. The number of trees needs to find a balance between model performance and computational efficiency. Too many trees can increase computational overhead (especially with large amounts of data), while too few trees may lead to suboptimal performance. When the number of decision trees is 50, the out-of-bag root mean square error ($RMSE^{out-of-bag}$) is 40.06 cm, while when the number of decision trees is 100, the $RMSE^{out-of-bag}$ is 39.05 cm (Table 4). The model performance has been stabilized when the decision tree is 100. Therefore, 100 decision trees are chosen for regression prediction.

The results of the random forest prediction model and interpolation model were compared (Table 5). The results show that the RMSE of the three interpolation models is not much different, but the random forest model has the smallest

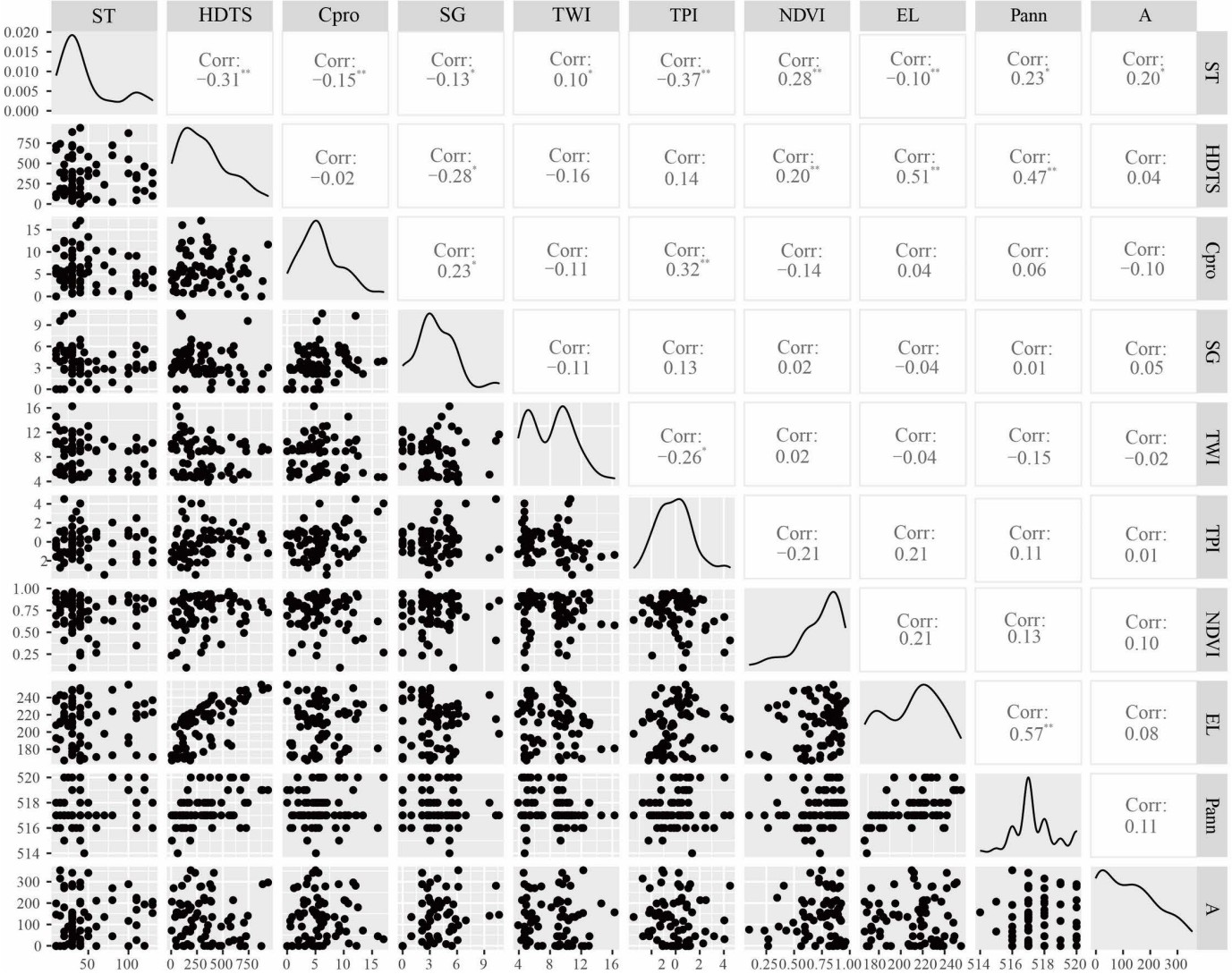

**Fig 3. Relationships between soil thickness and terrain variables.**

**Table 4. Out-of-bag data error of random forest.**

| Number of decision trees | RMSE$^{out-of-bag}$(cm) |
|---|---|
| 50 | 40.06 |
| 100 | 39.05 |

RMSE and the largest $R^2$. Scatter plots of observed and predicted values also show that the random forest results are closer to 1:1 curves, indicating better results than the spatial interpolation model (Fig 4). OOB errors are slightly better than cross-validation error results. In addition, nine environmental variables together explain about 57% of the spatial variability of black soil in the small watershed.

**Table 5. Comparison of estimated performance of random forest and interpolation models.**

| Model | RMSE(cm) | $R^2$ |
|---|---|---|
| Classical bayesian kriging | 43.55 | 0.13 |
| Ordinary kriging | 43.94 | 0.23 |
| Simple kriging | 42.09 | 0.12 |
| Random forest (OOB) | 36.01 | 0.57 |
| Random forest (CV) | 39.02 | 0.47 |

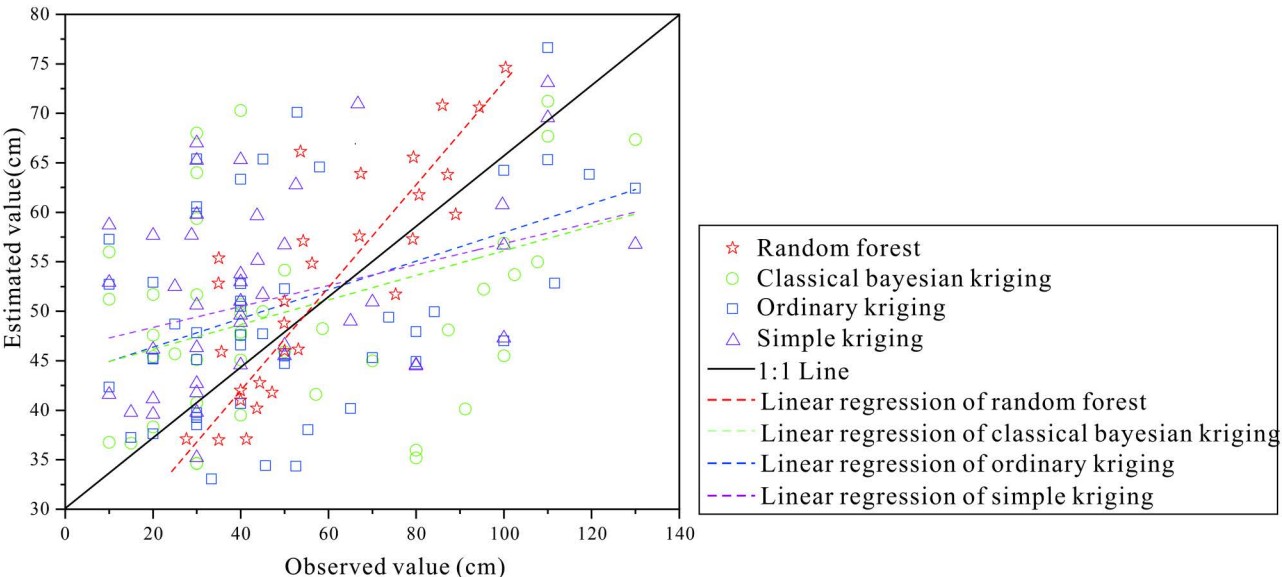

**Fig 4. Comparison of observed value and estimated value from random forest and interpolation models.**

## Variable importance

In the random forest model, the explanatory degree and importance of variables are important tools to evaluate the contribution of each variable in the model to the prediction results. The explanatory degree and importance of the nine variables have consistent changes in the prediction results of the thickness of black soil layer (Fig 5). EL has the greatest explanatory degree and importance. The interpretability and importance of TPI and NDVI are large and consistent. In addition, TWI and HDTS have explanations greater than 10%. $P_{ann}$, SG, CPro, and A are all less explanatory. CPro is as explanatory as A, but A is the least important of all variables.

## Spatial distribution of soil layer thickness

The interpolation model shows that the distribution of black soil in the small watershed is thick in the east and west and thin in the middle. The ordinary kriging prediction results can better reflect the difference in thickness than the other two kinds of kriging results. The results of random forest prediction can reflect the terrain characteristics, indicating that the thick black soil is mainly distributed in the gully area at the foot of the slope and the eastern part of the small watershed, and the thin black soil is distributed in the higher elevation of the slope, which is more consistent with the actual survey results and topographic changes. The results of Kriging interpolation and random forest prediction indicate that the black

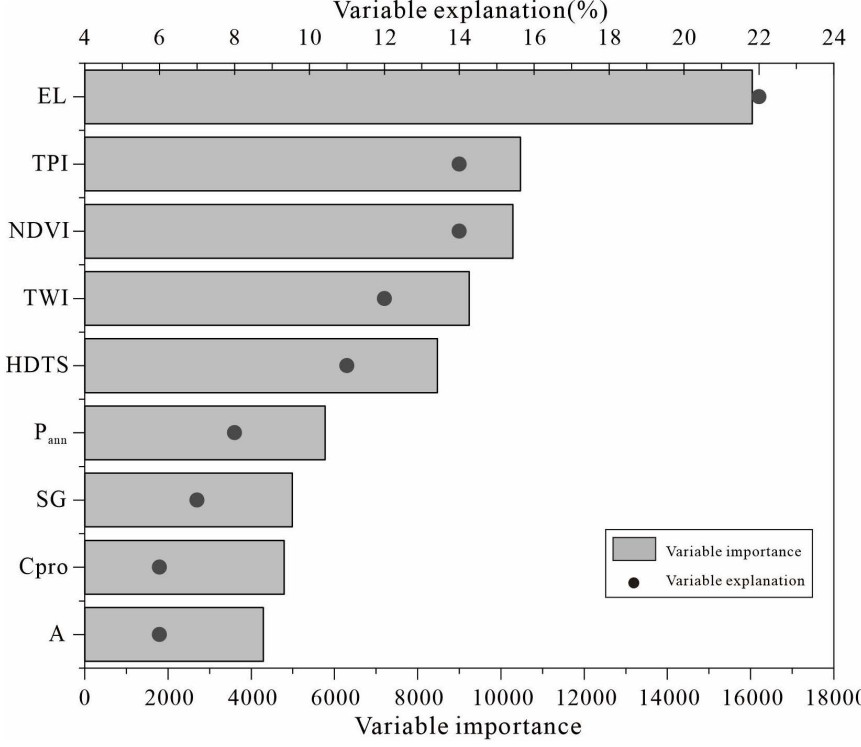

**Fig 5. Variable importance and variable explanation of random forest.**

soil layer in the east is thicker, which may be due to the lower elevation of the drainage area in the eastern part of the small watershed, and the overall thickness of the black soil is greater under the influence of water flow transport.

## Spatial distribution of erosion modulus

According to the factors in RUSLE determined above, the superposition calculation was carried out in the GIS platform, and the results were shown (Fig 7). The maximum erosion modulus of the small watershed in the study area is 1155.28 t/(km²·a), the average is 36.96 t/(km²·a), and the standard deviation is 53.82 t/(km²·a). The distribution of regional erosion modulus is extremely uneven. The area with small erosion modulus is distributed in the valley zone at low elevation. The erosion modulus is larger in areas with steep terrain and sparse vegetation.

## Discussion

### Comparison of random forest and interpolation model

The comparative analysis of black soil layer thickness prediction in the small watershed demonstrates the superior predictive performance of the random forest model over traditional interpolation approaches. As evidenced in Table 5, the random forest algorithm achieved significantly lower RMSE and higher R² values compared to the interpolation model. The spatial distribution patterns generated by random forest (Fig 6) better align with actual soil characteristics, suggesting enhanced prediction accuracy. This superiority likely stems from the effective noise mitigation capabilities and exceptional competence in resolving nonlinear relationships [43–45].In contrast, interpolation models require precise variogram construction involving complex parameter estimation processes that demand high-quality input data [46–48]. Their performance substantially deteriorates when handling limited or discontinuous datasets due to their inherent dependence on

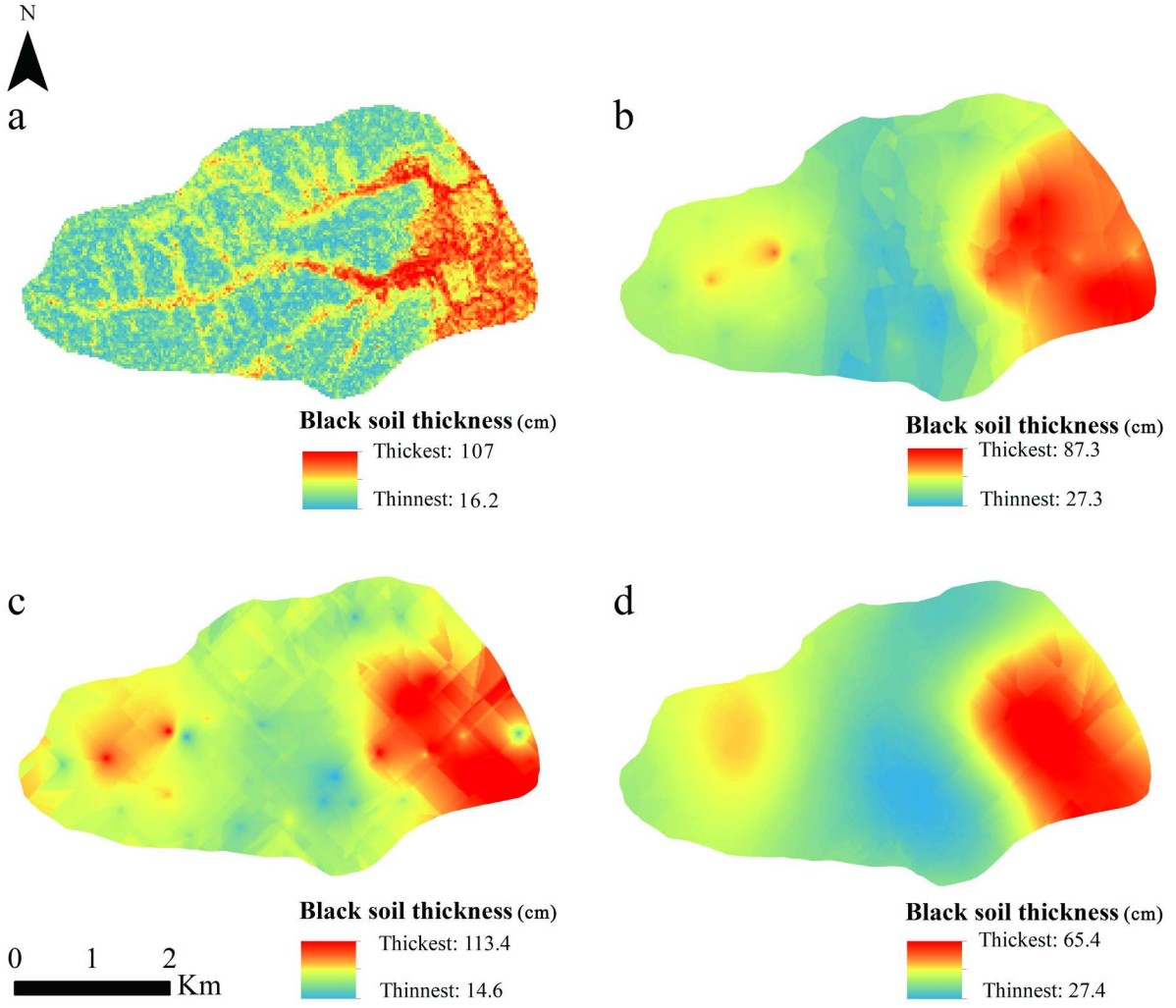

**Fig 6. Random forest and kriging modal results ((a) random forest; (b) classical bayesian kriging; (c), ordinary kriging; (d)simple kriging).**

spatial autocorrelation assumptions. The demonstrated effectiveness of random forest in this study highlights its particular advantage in processing complex, high-dimensional nonlinear spatial data [49–52], establishing it as a more robust methodological choice for soil layer thickness prediction. Li et al. [53] compared the sensitivity of 23 machine learning and spatial interpolation methods to input variables and the accuracy of prediction by using mud content samples from southwest Australia, and the results showed that random forest is the most robust regression prediction model. Chen et al. [10] used the stochastic forest model to predict the actual soil thickness according to the soil right censored data and extrapolated the model to the whole French mainland, with an accuracy of 79%-98% for the soil layer of 0.3 m to 2 m. Based on the measured soil $\delta^{15}N$ data, Zan et al. [54] established the optimal soil $\delta^{15}N$ relationship model for five climate zones, namely tropical, arid, temperate, boreal, and polar, using the regression algorithm of the random forest model and obtained satisfactory results ($R^2 = 0.68$, RMSE = 1.26‰). Large spatial variation of topography and geomorphology, less sample collection, and lower resolution are the main reasons why the interpolation model is inferior to random forest.

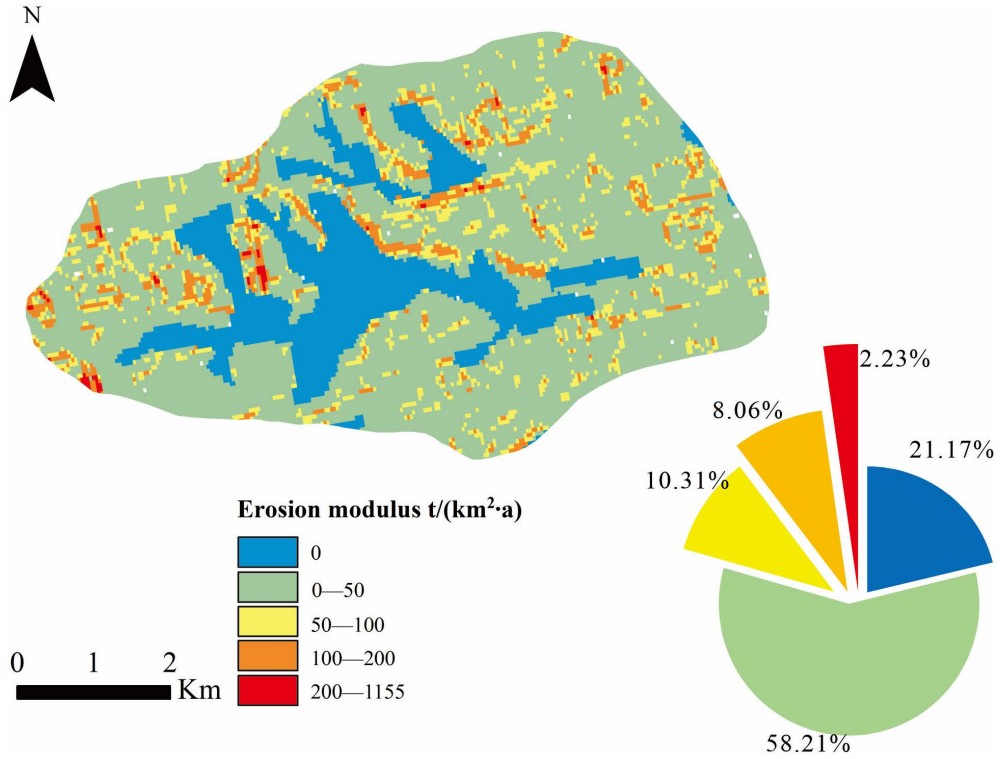

**Fig 7. RUSLE results grading and proportion.**

## The distribution characteristics and influencing factors of black soil layer thickness

The thickness distribution of the black soil in the small watershed is remarkably consistent with the topography and geomorphology. The black soil in the valley between the western slopes is thicker, the soil layers in the middle and top of the slopes are thinner, and the black soil is evenly distributed and thicker in the east. The average thickness of black soil predicted by random forest is 48.31 cm, which is consistent with the measured value of 48 cm. The predicted thickness of black soil layer range is 16.2 cm-107 cm, which is smaller than the measured range of 10 cm-130 cm, which may be affected by the smoothing effect of the random forest model. However, the smoothing effect may reduce the influence of random error and help to understand the spatial distribution pattern of the thickness of black soil layer [44].

TPI indicates the difference between the position of the terrain and the surrounding area. High TPI locations, such as ridges and hilltops, are susceptible to wind and water erosion. At the same time, the slope curvature in these areas is also large. Low TPI areas, such as valleys and slope foothills, tend to be deposit areas, close to the inter-slope streams, and the soil thickness is larger. Therefore, TPI, HDTS, and Cpro are significantly negatively correlated with soil thickness.

SG and ST also show a negative correlation, and soil particles in the larger slope will slide downward due to gravity, especially in the loose soil layer [14]. During rainfall, the fast flow of water easily washes away the surface soil. On steep slopes, the soil layer is thin, and vegetation and organisms struggle to survive, all of which slow down the development of soil.

Due to the high elevation, the soil thickness at the top of the slope is thin under the influence of wind and water erosion, so the soil thickness is negatively correlated with the altitude [55,56]. However, during the actual investigation, it was observed that the vegetation on the top of some slopes was dense and the soil layer was not thin, which may be affected by artificial afforestation. TWI reflects the characteristics of regional water accumulation. On flat or depressed terrain,

water is easy to accumulate, resulting in thicker soil [57]. NDVI, $P_{ann}$, and A all influence soil thickness through water and thermal conditions. In the northeastern region, south-facing slopes receive more sunlight, resulting in higher annual soil temperatures compared to north-facing slopes. These areas also have greater vegetation coverage. The northeastern region belongs to a temperate continental climate, where moisture brought by the southeast monsoon leads to more precipitation on south-facing slopes. Ample precipitation and active biological activity contribute to soil formation and development [58].

Terrain indirectly affects the spatial distribution of thickness of the black soil layer by influencing surface runoff, soil erosion, soil deposition, and soil temperature, especially in areas with steep slopes. Therefore, the spatial heterogeneity of soil thickness is the result of the interaction between soil processes (such as soil erosion and deposition) and surface runoff. Terrain parameters have varying degrees of importance in determining the thickness of black soil layer. In the study area, elevation is highly negatively correlated with soil thickness, profoundly influencing weathering processes and material transport in small watersheds. These factors are the most important determinants of the spatial pattern of soil thickness within the watershed. Particularly around the hilltops of slopes, an increase in elevation may accelerate soil erosion while soil deposition is enhanced in the valley areas. Li et al. [16] also found that elevation is the most important topographic parameter in small watersheds in southwestern China. TPI and HDTS are important topographic parameters influencing soil thickness [15,59]. NDVI is an important variable for soil estimation [8,56]. TWI and $P_{ann}$ influence erosion and deposition by controlling the movement of water and sediment [60,61]. SG and Cpro affect the acceleration of water flow and material movement, but their importance is relatively low, which is consistent with some earlier studies on soil depth estimation [15,56]. A has the lowest importance, and other topographic parameters likely play a more significant role in influencing changes in soil thickness.

## Soil erosion risk assessment

Soil thickness is an important physical property of the soil directly related to its ability to retain water and resist erosion [62]. The erosion modulus can be used to quantitatively analyze and predict the erosion degree of soil under specific conditions. A high erosion modulus indicates that soil in the region is vulnerable to erosion, while a low erosion modulus indicates that soil has good erosion resistance [63,64]. The combination of soil thickness and erosion modulus provides a more comprehensive consideration of the conditions under which soil erosion occurs. It breaks the limitation of a single indicator and comprehensively evaluates the interactive effects of external environmental factors and the characteristics of the soil itself. The results of random forest prediction of thickness of black soil layer were divided into four grades: grade I (> 60 cm), grade II (30 cm—60 cm), grade III (20 cm—30 cm), and grade IV (0–20 cm). The erosion modulus of the small watershed was divided into five grades: grade I (no erosion), grade II (slight erosion), grade III (mild erosion), grade IV (medium erosion), and grade V (severe erosion). According to the combination of the two, the risk of soil erosion in the small watershed is divided into four levels: no risk, low risk, medium risk, and high risk (Table 6).

Table 6. Soil erosion classification results.

| SEM<br>Soil thickness | I | II | III | IV | V |
|---|---|---|---|---|---|
| I | 1 | 1 | 1 | 2 | 3 |
| II | 1 | 1 | 2 | 2 | 3 |
| III | 2 | 2 | 2 | 3 | 4 |
| IV | 3 | 3 | 3 | 4 | 4 |

Note:1, no risk; 2, low risk; 3, medium risk; 4, high risk.

The proportion of the no-risk area is 21.91%, mainly distributed in the interfluve valleys and the gently sloping areas in the eastern part of small watersheds. The low-risk area accounts for 62.21% and is widely spread outside the no-risk area. The total area of the medium-risk and high-risk zones accounts for 15.88%, with island-like and strip-like distributions (Fig 8).

The no-risk area is primarily composed of soil accumulation zones. The low-risk area is mainly found in the sloped farming regions, covering a large area characterized by shallow erosion and surface erosion and significantly influenced by human agricultural activities. Terracing should be utilized, crop row directions should be changed, and vegetation strips should be planted to alter the microtopography, slope length, and gradient, thereby reducing erosion and mitigating ecological risks. The medium-risk and high-risk areas are located in regions with higher elevations, steeper slopes, and less vegetation, where shallow gully and incision erosion develop. Natural conditions and landform parameters are the controlling factors. These areas are small in size, with poor soil quality and low agricultural productivity. It is recommended to implement reforestation and grassland restoration, and deep eroded gullies should be filled and further subjected to biological restoration measures.

## Conclusion

Accurate estimation of the regional thickness of black soil layer and soil erosion risk is very important for land use, agricultural planting planning, and soil erosion monitoring and control. In this study, a soil profile survey was carried out in a small watershed in a typical black soil area. Combined with terrain landscape and vegetation climate parameters, random forest and interpolation models were selected to estimate the spatial thickness of black soil in the small watersheds, and combined with the RUSLE soil and water loss equation, soil and water erosion risk assessment was carried out on small watersheds. The results showed that:

(1)  Compared with the interpolation model, the random forest model has a smaller RMSE (34.05 cm) and larger $R^2$ (0.57), which is more suitable for processing discontinuous and nonlinear high-dimensional data.

(2)  The thickness of the predicted black soil ranges from 16.2 cm to 107 cm, and the average thickness of the predicted black soil in the small watershed is 48.31 cm, which is consistent with the measured value of 48 cm. Among the nine

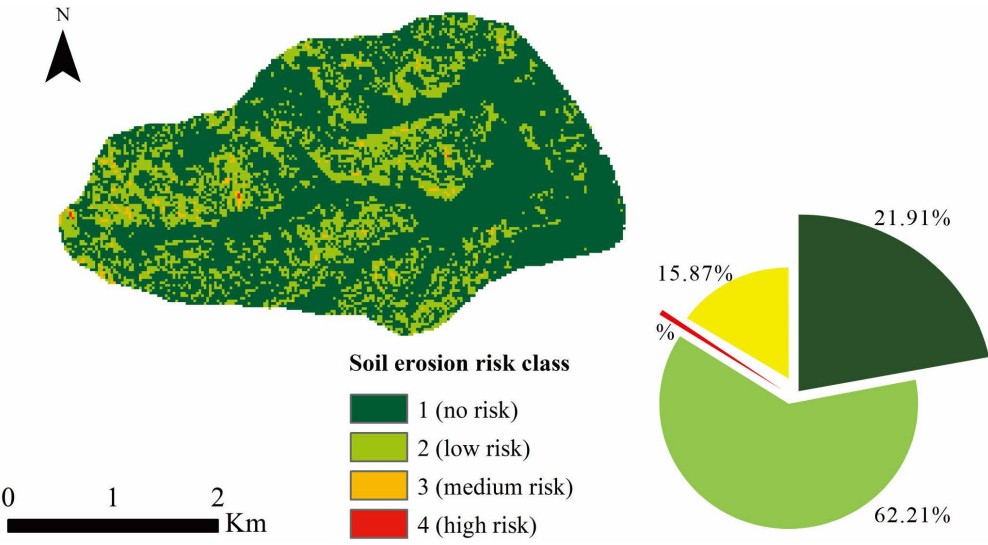

**Fig 8.  Risk assessment levels and proportions of soil and water erosion.**

environmental parameters, elevation is the most important parameter that profoundly affects the growth and development of soil.

(3) The risk of soil erosion in the small watershed is divided into 4 levels according to no risk, low risk, medium risk, and high risk. The Level 1 area, consisting of soil accumulation zones, accounts for 21.91%. The Level 2 area, which accounts for 62.21%, is mainly found in sloped farmland, where the primary management measures include terracing, changing crop row directions, and planting vegetation strips along field ridges. The Level 3 and Level 4 areas together account for 15.88%, with a strip-like and island-like distribution. These areas should be converted from farmland to forest or grassland, and deep eroded gullies should be managed with a combination of engineering and biological measures.

## Author contributions

**Formal analysis:** Kai Liu.

**Funding acquisition:** Huimin Dai, Kai Liu.

**Investigation:** Keke Xu, Huimin Dai, Kai Liu, Guanxin Du.

**Methodology:** Keke Xu.

**Resources:** Keke Xu.

**Software:** Chaoqun Chen.

**Writing – original draft:** Keke Xu.

**Writing – review & editing:** Keke Xu, Xujiao Zhang, Kai Liu, Cheng Qian.

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
