## [Decision Letter · Decision Letter 0]

13 Jan 2025

PONE-D-24-57959Black soil thickness prediction and soil erosion risk assessment in a small watershed in Northeast ChinaPLOS ONE

Dear Dr. xu,

Thank you for submitting your manuscript to PLOS ONE. After careful consideration, we feel that it has merit but does not fully meet PLOS ONE’s publication criteria as it currently stands. Therefore, we invite you to submit a revised version of the manuscript that addresses the points raised during the review process.

We look forward to receiving your revised manuscript.

Kind regards,

Sher Muhammad, PhD

Academic Editor

PLOS ONE

Journal Requirements:

1. When submitting your revision, we need you to address these additional requirements. Please ensure that your manuscript meets PLOS ONE's style requirements, including those for file naming. The PLOS ONE style templates can be found at https://journals.plos.org/plosone/s/file?id=wjVg/PLOSOne_formatting_sample_main_body.pdf and https://journals.plos.org/plosone/s/file?id=ba62/PLOSOne_formatting_sample_title_authors_affiliations.pdf 2. We note that the grant information you provided in the ‘Funding Information’ and ‘Financial Disclosure’ sections do not match.  When you resubmit, please ensure that you provide the correct grant numbers for the awards you received for your study in the ‘Funding Information’ section. 3. In the online submission form you indicate that your data is not available for proprietary reasons and have provided a contact point for accessing this data. Please note that your current contact point is a co-author on this manuscript. According to our Data Policy, the contact point must not be an author on the manuscript and must be an institutional contact, ideally not an individual. Please revise your data statement to a non-author institutional point of contact, such as a data access or ethics committee, and send this to us via return email. Please also include contact information for the third party organization, and please include the full citation of where the data can be found. 4. PLOS requires an ORCID iD for the corresponding author in Editorial Manager on papers submitted after December 6th, 2016. Please ensure that you have an ORCID iD and that it is validated in Editorial Manager. To do this, go to ‘Update my Information’ (in the upper left-hand corner of the main menu), and click on the Fetch/Validate link next to the ORCID field. This will take you to the ORCID site and allow you to create a new iD or authenticate a pre-existing iD in Editorial Manager. 5. We note that Figure 1 in your submission contain [map/satellite] images which may be copyrighted. All PLOS content is published under the Creative Commons Attribution License (CC BY 4.0), which means that the manuscript, images, and Supporting Information files will be freely available online, and any third party is permitted to access, download, copy, distribute, and use these materials in any way, even commercially, with proper attribution. For these reasons, we cannot publish previously copyrighted maps or satellite images created using proprietary data, such as Google software (Google Maps, Street View, and Earth). For more information, see our copyright guidelines: http://journals.plos.org/plosone/s/licenses-and-copyright. We require you to either (1) present written permission from the copyright holder to publish these figures specifically under the CC BY 4.0 license, or (2) remove the figures from your submission: 1. You may seek permission from the original copyright holder of Figure 1 to publish the content specifically under the CC BY 4.0 license.   We recommend that you contact the original copyright holder with the Content Permission Form (http://journals.plos.org/plosone/s/file?id=7c09/content-permission-form.pdf) and the following text:“I request permission for the open-access journal PLOS ONE to publish XXX under the Creative Commons Attribution License (CCAL) CC BY 4.0 (http://creativecommons.org/licenses/by/4.0/). Please be aware that this license allows unrestricted use and distribution, even commercially, by third parties. Please reply and provide explicit written permission to publish XXX under a CC BY license and complete the attached form.” Please upload the completed Content Permission Form or other proof of granted permissions as an ""Other"" file with your submission. In the figure caption of the copyrighted figure, please include the following text: “Reprinted from [ref] under a CC BY license, with permission from [name of publisher], original copyright [original copyright year].” 2. If you are unable to obtain permission from the original copyright holder to publish these figures under the CC BY 4.0 license or if the copyright holder’s requirements are incompatible with the CC BY 4.0 license, please either i) remove the figure or ii) supply a replacement figure that complies with the CC BY 4.0 license. Please check copyright information on all replacement figures and update the figure caption with source information. If applicable, please specify in the figure caption text when a figure is similar but not identical to the original image and is therefore for illustrative purposes only.The following resources for replacing copyrighted map figures may be helpful: USGS National Map Viewer (public domain): http://viewer.nationalmap.gov/viewer/The Gateway to Astronaut Photography of Earth (public domain): http://eol.jsc.nasa.gov/sseop/clickmap/Maps at the CIA (public domain): https://www.cia.gov/library/publications/the-world-factbook/index.html and https://www.cia.gov/library/publications/cia-maps-publications/index.htmlNASA Earth Observatory (public domain): http://earthobservatory.nasa.gov/Landsat:
http://landsat.visibleearth.nasa.gov/USGS EROS (Earth Resources Observatory and Science (EROS) Center) (public domain): http://eros.usgs.gov/#Natural Earth (public domain): http://www.naturalearthdata.com/

Additional Editor Comments:

Two anonymous reviewers have reviewed the manuscript. The manuscript requires substantial revisions to address the issues in the present form. The Random Forest method's validation on a smaller range of observed values (30–60) as opposed to the entire range (10–130) covered by the kriging methods is one of the main concerns brought up by the reviewers. This disparity calls into question both the stated high R2 and the validity of the Random Forest results. To guarantee reliable comparisons, this approach must be tested across the entire spectrum. Furthermore, the wording "using raster calculations" in Table 1's description of precipitation data interpolation is imprecise. The paper should discuss the drawbacks of integrating coarse-resolution data with pixel sizes of 30 m and make clear how interpolation techniques affect the results. A comprehensive language evaluation is advised since the language has to be improved. The overall impact is limited by the study's unclear combination of soil erosion risk assessment and mapping the thickness of the black soil layer. A more unified process can result from including soil erosion as a variable in thickness mapping. The title has to be clarified, the units in the figures need to be formatted correctly, the novelty of this work needs to be better discussed with references to other research, and the sample approach needs to be clearly defined.

Reviewers' comments:

Reviewer's Responses to Questions

**Comments to the Author**

1. Is the manuscript technically sound, and do the data support the conclusions?

Reviewer #1: No

Reviewer #2: Yes

2. Has the statistical analysis been performed appropriately and rigorously? 

Reviewer #1: No

Reviewer #2: Yes

3. Have the authors made all data underlying the findings in their manuscript fully available?

Reviewer #1: No

Reviewer #2: Yes

4. Is the manuscript presented in an intelligible fashion and written in standard English?

Reviewer #1: No

Reviewer #2: No

5. Review Comments to the Author

Reviewer #1: Thank you for the opportunity to review this research. The topic is important and the research approach is good, but I have some concerns about the execution and communication. I hope this can be revised to reach a broader audience though probably not in this journal.

My main concern pertains to Figure 4: All of the kriging methods show data points with observed values ranging from 10 to 130, matching the range of values reported in the paper. However, the Random Forest method only shows observed values ranging from about 30 to 60, and the linear regression is based on this small range. I don't think the overall results and much larger R^2 can be reliable if the Random Forest method is not tested against all of the observed values.

A secondary concern pertains to the methods used to interpolate the precipitation data listed in Table 1. The methods section should say more than "using raster calculations" (as on lines 139 and 142) and should discuss the relative accuracy or error inherent in combining data from much coarser resolutions with the 30m pixel size used in the study.

While the English usage is mostly understandable, there are minor errors throughout the paper that should be addressed. For example (not a complete list):

Check throughout the paper for the proper use of English articles (a, an, the): _the_ hydrologic cycle, _the_ Random Forest method, etc.

At line 45, please check the units and superscripts on line 45 (10^6 not 106, km^2 or ha?).

At line 108, what is "soil erosion, soil erosion ditch" (possibly gully formation?).

At line 116, rephrase the survey site description in better English.

At line 149, say Methods not Method and add a short introductory paragraph for this section.

At line 227, you say that the number of trees has a significant impact, but Table 4 shows that doubling from 50 to 100 trees only reduces the RMSE^oob by <2%.

In Figures 2 and 6, make sure units are given in parenthesis, not after a slash (which can mean "per"): (cm) not /cm, (%) not /%

Reviewer #2: This manuscript compared several models for predicting the thickness of black soil layer in a small watershed in Northeast China. Authors also explored the soil erosion risk using RUSLE model in the study area. Authors found that the random forest model outperformed the kriging models, with smaller RMSE and larger R². Soil erosion risk assessment revealed that areas with no risk and low risk accounted for 21.91% and 62.21%, respectively. The investigated topic is interesting and important for evaluating and protecting black soil in China. Nevertheless, this work has several major limitations: (1) the work on thickness of black soil layer mapping and soil erosion risk assessment has no linkage. The outcome of thickness map of black soil layer does not provide any useful information on soil erosion risk assessment in the current workflow. However, if you put soil erosion risk assessment as a starting point, then adding this variable as an input for thickness mapping, which is more logical. (2) the current knowledge has been poorly described, limiting the novelty of this work; (3) Please provide both out-of-bag validation and k-fold cross-validation results; (4) there are many mistakes regarding the grammars and duplicated sentences. Please carefully check the writing quality before the resubmission. In summary, this manuscript is not suitable to be published now, but can be reconsidered after re-submission.

Detailed comments:

Title: The use of “Black soil thickness” is confusing. Please consider the use of thickness of black soil layer.

Line 29: kriging methods (classical, ordinary, and simple) have been rarely used nowadays.

Lines 30-31: It is not clear how to connect the work on mapping thickness of black soil layer and soil erosion assessment.

Line 32: Please provide the values for RMSE and R2.

Line 45: 106 km2, 6 should be in superscript. Please use km2 for arable land as well.

Line 52: It is important to describe the definition of black soil layer. For most readers outside of China, it is not easy to understand the exact meaning.

Lines 57-60: As you stated here, soil erosion plays an important in shaping the thickness of black soil layer, so why not including soil erosion as an input for mapping thickness of black soil layer?

Line 61: It would be important to start the paragraph with the fact that soil thickness is the worst predicted soil information as summarized in the review by Chen et al. (2022). Therefore, there is a need to develop new methodology for soil thickness mapping.

Chen, S., Arrouays, D., Mulder, V.L., Poggio, L., Minasny, B., Roudier, P., Libohova, Z., Lagacherie, P., Shi, Z., Hannam, J. and Meersmans, J., 2022. Digital mapping of GlobalSoilMap soil properties at a broad scale: A review. Geoderma, 409, p.115567.

Lines 78-81: It is important to acknowledge previous studies on using machine learning model to predict soil thickness. Some relevant references are listed below, and you are explore more other interesting work as well:

Malone, B. and Searle, R., 2020. Improvements to the Australian national soil thickness map using an integrated data mining approach. Geoderma, 377, p.114579.

Liu, F., Zhang, G.L., Song, X., Li, D., Zhao, Y., Yang, J., Wu, H. and Yang, F., 2020. Liu, F., Wu, H., Zhao, Y., Li, D., Yang, J.L., Song, X., Shi, Z., Zhu, A.X. and Zhang, G.L., 2022. Mapping high resolution national soil information grids of China. Science Bulletin, 67(3), pp.328-340.

Lines 86-90: The current knowledge is missing before stating the objectives.

Line 102: N and E are missing after the coordinates.

Lines 114-123: Please specify the sampling design of the four survey routes.

Line 134: (1) for the equation.

Lines 137-142: Two sentences are replicated.

Line 148: In Table 1, why not using the National Soil Information Grids of China at 90 m resolution?

Lines 153-154: 500 trees would be more stable.

Line 167: In Figure 1, authors indicated the sites for model testing, but here you mentioned that OOB error was used. This is confusing.

Since the results would be changed if authors using the soil erosion map as an input. I stop here and would be happy to provide more comments in the next round.

6. PLOS authors have the option to publish the peer review history of their article (what does this mean? ). If published, this will include your full peer review and any attached files.

**Do you want your identity to be public for this peer review?** For information about this choice, including consent withdrawal, please see our Privacy Policy .

Reviewer #1: No

Reviewer #2: No

---

## [Author Response · Author response to Decision Letter 1]

19 Mar 2025

The format of the article has been corrected according to the formatting requirements of your journal.

Answer Revisions have been made to ‘Funding Information’ and ‘Financial Disclosure’.

3. In the online submission form you indicate that your data is not available for proprietary reasons and have provided a contact point for accessing this data. Please note that your current contact point is a co-author on this manuscript. According to our Data Policy, the contact point must not be an author on the manuscript and must be an institutional contact, ideally not an individual. Please revise your data statement to a non-author institutional point of contact, such as a data access or ethics committee, and send this to us via return email. Please also include contact information for the third party organization, and please include the full citation of where the data can be found.

Answer: The contact person for data is the Key Laboratory of Blackland Evolution and Ecological Effects, Shenyang Geological Survey Centre, China Geological Survey. Tel: +86-024-86002998. Email: cgssyzx_data@yeah.net

Answer Modified in the system

Answer Figure 1 contains [map/satellite] images from NASA platforms. This data is free for use, modification, and distribution, including academic publication, in accordance with NASA's open data services and software strategy. Data sources have been cited in the text . Open Data, Services, and Software Policies | NASA Earthdata

6. Additional Editor Comments:

Two anonymous reviewers have reviewed the manuscript. The manuscript requires substantial revisions to address the issues in the present form. The Random Forest method's validation on a smaller range of observed values (30–60) as opposed to the entire range (10–130) covered by the kriging methods is one of the main concerns brought up by the reviewers. This disparity calls into question both the stated high R2 and the validity of the Random Forest results. To guarantee reliable comparisons, this approach must be tested across the entire spectrum. Furthermore, the wording "using raster calculations" in Table 1's description of precipitation data interpolation is imprecise. The paper should discuss the drawbacks of integrating coarse-resolution data with pixel sizes of 30 m and make clear how interpolation techniques affect the results. A comprehensive language evaluation is advised since the language has to be improved. The overall impact is limited by the study's unclear combination of soil erosion risk assessment and mapping the thickness of the black soil layer. A more unified process can result from including soil erosion as a variable in thickness mapping. The title has to be clarified, the units in the figures need to be formatted correctly, the novelty of this work needs to be better discussed with references to other research, and the sample approach needs to be clearly defined.

The Random Forest method's validation on a smaller range of observed values (30–60) as opposed to the entire range (10–130) covered by the kriging methods is one of the main concerns brought up by the reviewers. This disparity calls into question both the stated high R2 and the validity of the Random Forest results. To guarantee reliable comparisons, this approach must be tested across the entire spectrum.

Answer In the previous study due to the non-normality of the black soil layer thickness data, the random forest model was over-sampled, resulting in an exaggerated R2 (0.79). This time, overfitting was prevented by limiting the depth of the random forest decision tree, and an R2 of 0.57 was obtained.

Furthermore, the wording "using raster calculations" in Table 1's description of precipitation data interpolation is imprecise. The paper should discuss the drawbacks of integrating coarse-resolution data with pixel sizes of 30 m and make clear how interpolation techniques affect the results.

Answer The monthly precipitation data in Table 1 were rastered to obtain annual average precipitation data. The precipitation and dem data were then resampled to 30m resolution. Considering that the data are continuous and resampled using the averaging method, although the process loses some data precision, the margin of error is known to be sufficiently small so that no further errors are reported.

The overall impact is limited by the study's unclear combination of soil erosion risk assessment and mapping the thickness of the black soil layer. A more unified process can result from including soil erosion as a variable in thickness mapping.

Answer� Soil thickness, as an intrinsic physical property, plays a pivotal role in determining soil stability, water retention capacity, and erosion resistance. Particularly in regions characterized by intense precipitation or steep topographic gradients, thicker soil profiles demonstrate enhanced resistance to erosional processes, effectively mitigating the impact of external erosive forces. Conventional soil erosion assessment predominantly relies on a unidimensional approach utilizing the soil erosion modulus, which solely quantifies erosion intensity per unit time. However, the erosion modulus and soil thickness were combined to create an integrated two-dimensional assessment model of “erosion intensity - soil resistance”. enabling more accurate soil erosion risk assessment. This innovative model classifies erosion risk into four distinct categories: (1) high-risk areas characterized by high erosion modulus coupled with thin soil layers; (2) medium-risk areas exhibiting high erosion modulus with thick soil layers; (3) low-risk areas featuring low erosion modulus and thin soil layers; and (4) risk-free areas demonstrating both low erosion modulus and substantial soil thickness.

When the random forest model encounters highly correlated features, it may "overfit" certain variables, leading to a decrease in regression accuracy. The erosion modulus calculated by RUSLE has a strong linear or nonlinear correlation with input variables of the random forest model (such as slope, vegetation index, elevation, precipitation, etc.). Therefore, when it is included in the random forest model, it may lead to information redundancy, which can affect the model's performance. As a result, the RUSLE results were not used as input variables for predicting soil thickness.

The title has to be clarified, the units in the figures need to be formatted correctly.

Answer The title has been revised in accordance with the review comments and the formatting of the units in the text has been revised.

The novelty of this work needs to be better discussed with references to other research.

Answer Recent research advances have been added and compared to the present study.

The sample approach needs to be clearly defined

Answer Route setting principles and black soil layer definitions have been added so that the article can be easily understood. 

Reviewer #1: Thank you for the opportunity to review this research. The topic is important and the research approach is good, but I have some concerns about the execution and communication. I hope this can be revised to reach a broader audience though probably not in this journal.

My main concern pertains to Figure 4: All of the kriging methods show data points with observed values ranging from 10 to 130, matching the range of values reported in the paper. However, the Random Forest method only shows observed values ranging from about 30 to 60, and the linear regression is based on this small range. I don't think the overall results and much larger R2 can be reliable if the Random Forest method is not tested against all of the observed values.

Answer: Bias in the black soil thickness data in the previous model resulted in an exaggerated R2 (0.79). The R2 was now changed to 0.57 by limiting the depth of the decision tree to prevent overfitting.

A secondary concern pertains to the methods used to interpolate the precipitation data listed in Table 1. The methods section should say more than "using raster calculations" (as on lines 139 and 142) and should discuss the relative accuracy or error inherent in combining data from much coarser resolutions with the 30m pixel size used in the study.

Answer: The average monthly precipitation is added together to get the average annual precipitation. The annual average precipitation data were interpolated to a 30 m resolution using geostatistical methods. Due to the small size of the study area and limited spatial variability in precipitation, errors associated with data resolution conversion were not explicitly considered. Lines 139 to 142 are descriptions of Topographic parameters calculated using 30m resolution data, so no relative accuracy or error is involved

While the English usage is mostly understandable, there are minor errors throughout the paper that should be addressed. For example (not a complete list):

Check throughout the paper for the proper use of English articles (a, an, the): _the_ hydrologic cycle, _the_ Random Forest method, etc.

Answer: Has been rechecked and modified

At line 45, please check the units and superscripts on line 45 (10^6 not 106, km^2 or ha?).

Answer Have been modified

At line 108, what is "soil erosion, soil erosion ditch" (possibly gully formation?).

Answer Modified as soil erosion and gully

At line 116, rephrase the survey site description in better English.

Answer Have been modified

At line 149, say Methods not Method and add a short introductory paragraph for this section.

Answer Have been modified

At line 227, you say that the number of trees has a significant impact, but Table 4 shows that doubling from 50 to 100 trees only reduces the RMSE^oob by <2%.

In Figures 2 and 6, make sure units are given in parenthesis, not after a slash (which can mean "per"): (cm) not /cm, (%) not /%

Answer Have been modified

Reviewer #2: This manuscript compared several models for predicting the thickness of black soil layer in a small watershed in Northeast China. Authors also explored the soil erosion risk using RUSLE model in the study area. Authors found that the random forest model outperformed the kriging models, with smaller RMSE and larger R². Soil erosion risk assessment revealed that areas with no risk and low risk accounted for 21.91% and 62.21%, respectively. The investigated topic is interesting and important for evaluating and protecting black soil in China. Nevertheless, this work has several major limitations: (1) the work on thickness of black soil layer mapping and soil erosion risk assessment has no linkage. The outcome of thickness map of black soil layer does not provide any useful information on soil erosion risk assessment in the current workflow. However, if you put soil erosion risk assessment as a starting point, then adding this variable as an input for thickness mapping, which is more logical. (2) the current knowledge has been poorly described, limiting the novelty of this work; (3) Please provide both out-of-bag validation and k-fold cross-validation results; (4) there are many mistakes regarding the grammars and duplicated sentences. Please carefully check the writing quality before the resubmission. In summary, this manuscript is not suitable to be published now, but can be reconsidered after re-submission.

Detailed comments:

Title: The use of “Black soil thickness” is confusing. Please consider the use of thickness of black soil layer.

Answer�thickness of black soil layer is more accuracy. Have been modified

Line 29: kriging methods (classical, ordinary, and simple) have been rarely used nowadays.

Answer With the emergence of new interpolation methods, the use of methods like classical, ordinary and simple Kriging has indeed decreased, but this does not mean that they are no longer valuable. The Kriging method is still an effective tool in the field of geostatistics. Therefore, random forest and Kriging are selected to compare the robustness of their models

Lines 30-31: It is not clear how to connect the work on mapping thickness of black soil layer and soil erosion assessment.

Answer� RUSLE (Revised Universal Soil Loss Equation) model has the characteristics of simple structure, easy parameter acquisition, simple calculation and high precision. It has been widely used in soil erosion modulus estimation at home and abroad. The erosion modulus mainly reflects the influence of erosion factors such as precipitation, slope, and vegetation coverage on soil erosion. These factors drive soil loss by external forcing. Soil thickness, as an intrinsic physical property, plays a pivotal role in determining soil stability, water retention capacity, and erosion resistance. Particularly in regions characterized by intense precipitation or steep topographic gradients, thicker soil profiles demonstrate enhanced resistance to erosional processes, effectively mitigating the impact of external erosive forces. Conventional soil erosion assessment predominantly relies on a unidimensional approach utilizing the soil erosion modulus, which solely quantifies erosion intensity per unit time. However, by integrating soil thickness parameters, a comprehensive two-dimensional evaluation framework of "erosion intensity versus soil layer resistance" can be established, enabling more accurate soil erosion risk assessment. This innovative model classifies erosion risk into four distinct categories: (1) high-risk areas characterized by high erosion modulus coupled with thin soil layers; (2) medium-risk areas exhibiting high erosion modulus with thick soil layers; (3) low-risk areas featuring low erosion modulus and thin soil layers; and (4) risk-free areas demonstrating both low erosion modulus and substantial soil thickness.

Line 32: Please provide the values for RMSE and R2.

Answer Have been modified

Line 45: 106 km2, 6 should be in superscript. Please use km2 for arable land as well.

Answer Have been modified

Line 52: It is important to describe the definition of black soil layer. For most readers outside of China, it is not easy to understand the exact meaning.

Answer Description and definition of the black soil layer have been added.

Lines 57-60: As you stated here, soil erosion plays an important in shaping the thickness of black soil layer, so why not including soil erosion as an inp

---

## [Decision Letter · Decision Letter 1]

1 Apr 2025

PONE-D-24-57959R1Black soil layer thickness prediction and soil erosion risk assessment in a small watershed in Northeast ChinaPLOS ONE

Dear Dr. liu,

Thank you for submitting your manuscript to PLOS ONE. After careful consideration, we feel that it has merit but does not fully meet PLOS ONE’s publication criteria as it currently stands. Therefore, we invite you to submit a revised version of the manuscript that addresses the points raised during the review process.

We look forward to receiving your revised manuscript.

Kind regards,

Sher Muhammad, PhD

Academic Editor

PLOS ONE

Journal Requirements:

Additional Editor Comments:

Dear Authors, Thank you for the revisions. The revised version has been reviewed by two reviewers, recommending a major revision before further consideration.

You are encouraged to address each of the reviewers’ comments and resubmit the revision. We are looking forward to receive your revised manuscript.

Reviewers' comments:

Reviewer's Responses to Questions

**Comments to the Author**

1. If the authors have adequately addressed your comments raised in a previous round of review and you feel that this manuscript is now acceptable for publication, you may indicate that here to bypass the “Comments to the Author” section, enter your conflict of interest statement in the “Confidential to Editor” section, and submit your "Accept" recommendation.

Reviewer #1: All comments have been addressed

Reviewer #2: (No Response)

2. Is the manuscript technically sound, and do the data support the conclusions?

Reviewer #1: Yes

Reviewer #2: (No Response)

3. Has the statistical analysis been performed appropriately and rigorously? 

Reviewer #1: Yes

Reviewer #2: (No Response)

4. Have the authors made all data underlying the findings in their manuscript fully available?

Reviewer #1: Yes

Reviewer #2: (No Response)

5. Is the manuscript presented in an intelligible fashion and written in standard English?

Reviewer #1: Yes

Reviewer #2: (No Response)

6. Review Comments to the Author

Reviewer #1: (No Response)

Reviewer #2: This revised manuscript has been improved after addressing my previous comments, while many issues still need to be revised.

(1) the grammar should be greatly improved. Some examples are listed below: “Use the RUSLE erosion equation and soil thickness to create a soil erosion rating scale that integrates external factors and the soil's properties”, “The topography of the study area is high in the west and low in the east, high in the south and north, and low in the center.”.

(2) The reference 10 is not a review paper. Please have a check.

(3) Lines 104-110: The part is wordy, you can shorten it.

(4) Line 88: soil-forming environment should be corrected as soil-forming environmental covariates.

(5) Line 89: Malong is a typo. It should be corrected as Malone.

(6) Line 92: quantile random forests should be corrected as QRF since it has been defined in in Line 87.

(7) Line 131: Fig.1 not Fig 1. Please check similar typos in the manuscript.

(8) Line 183: There is no need to define RUSLE again.

(9) Lines 203-215: The problem is that OOB is for random forest, so how you applied for Kriging interpolation. Please make it clear.

(10) Table 5: R2 should be corrected as R2.

(11) You have two c in the figure caption.

(12) Line 307: Why used the section named “RUSLE result” here? Please check the format throughout the manuscript.

7. PLOS authors have the option to publish the peer review history of their article (what does this mean? ). If published, this will include your full peer review and any attached files.

**Do you want your identity to be public for this peer review?** For information about this choice, including consent withdrawal, please see our Privacy Policy .

Reviewer #1: No

Reviewer #2: No

---

## [Author Response · Author response to Decision Letter 2]

11 Apr 2025

(1) the grammar should be greatly improved. Some examples are listed below: “Use the RUSLE erosion equation and soil thickness to create a soil erosion rating scale that integrates external factors and the soil's properties”, “The topography of the study area is high in the west and low in the east, high in the south and north, and low in the center.”.

The grammar of the entire text has been checked and modified.

(2) The reference 10 is not a review paper. Please have a check.

Modified to correct reference.

(3) Lines 104-110: The part is wordy, you can shorten it.

The sentence have been revised to“However, integrating soil thickness parameters establishes a two-dimensional "erosion intensity vs. soil resistance" framework, enabling precise erosion risk classification: (1) high-risk (high erosion modulus, thin soil layers); (2) medium-risk (high modulus, thick layers); (3) low-risk (low modulus, thin layers); and (4) risk-free (low modulus, thick layers).”

(4) Line 88: soil-forming environment should be corrected as soil-forming environmental covariates.

Have been modified.

(5) Line 89: Malong is a typo. It should be corrected as Malone.

Have been modified.

(6) Line 92: quantile random forests should be corrected as QRF since it has been defined in in Line 87.

Have been modified.

(7) Line 131: Fig.1 not Fig 1. Please check similar typos in the manuscript.

Have been modified and checked.

(8) Line 183: There is no need to define RUSLE again.

Have been modified.

(9) Lines 203-215: The problem is that OOB is for random forest, so how you applied for Kriging interpolation. Please make it clear.

OOB is just for random forest. Cross-validation was applied for Kriging interpolation. We also calculate random forest cross-validation to compare with Kriging interpolation.

(10) Table 5: R2 should be corrected as R2.

Have been modified.

(11) You have two c in the figure caption.

Have been modified.

(12) Line 307: Why used the section named “RUSLE result” here? Please check the format throughout the manuscript.

“RUSLE result” have been modified to “Spatial distribution of erosion modulus ”.

---

## [Decision Letter · Decision Letter 2]

24 Apr 2025

Black soil layer thickness prediction and soil erosion risk assessment in a small watershed in Northeast China

PONE-D-24-57959R2

Dear Dr. liu,

We’re pleased to inform you that your manuscript has been judged scientifically suitable for publication and will be formally accepted for publication once it meets all outstanding technical requirements.

Kind regards,

Dinesh Panday

Academic Editor

PLOS ONE

Additional Editor Comments (optional):

Thank you for your thorough revisions. There is a minor suggestion from the reviewer, please address it before sending to production unit.

Reviewers' comments:

Reviewer's Responses to Questions

**Comments to the Author**

1. If the authors have adequately addressed your comments raised in a previous round of review and you feel that this manuscript is now acceptable for publication, you may indicate that here to bypass the “Comments to the Author” section, enter your conflict of interest statement in the “Confidential to Editor” section, and submit your "Accept" recommendation.

Reviewer #1: All comments have been addressed

Reviewer #2: All comments have been addressed

2. Is the manuscript technically sound, and do the data support the conclusions?

Reviewer #1: Yes

Reviewer #2: Yes

3. Has the statistical analysis been performed appropriately and rigorously? 

Reviewer #1: Yes

Reviewer #2: Yes

4. Have the authors made all data underlying the findings in their manuscript fully available?

Reviewer #1: Yes

Reviewer #2: Yes

5. Is the manuscript presented in an intelligible fashion and written in standard English?

Reviewer #1: Yes

Reviewer #2: Yes

6. Review Comments to the Author

Reviewer #1: Thank you for the opportunity to review this revised paper. The new version adequately addresses the prior reviewer comments and should be published.

If there is an opportunity for a very minor revision, please explain or spell out the abbreviation SEM (soil erosion modulus) in Table 6; ideally, the SEM and thickness categories would also be defined in the figure row/column headings or in the figure caption, not just in the text of the paper.

Reviewer #2: All the suggestions have been well addressed in the revised manuscript, and therefore it can be accepted for publication.

7. PLOS authors have the option to publish the peer review history of their article (what does this mean? ). If published, this will include your full peer review and any attached files.

**Do you want your identity to be public for this peer review?** For information about this choice, including consent withdrawal, please see our Privacy Policy .

Reviewer #1: No

Reviewer #2: No

---

## [Editor Report · Acceptance letter]

PONE-D-24-57959R2

PLOS ONE

Dear Dr. liu,

I'm pleased to inform you that your manuscript has been deemed suitable for publication in PLOS ONE. Congratulations! Your manuscript is now being handed over to our production team.

Kind regards,

on behalf of

Dr. Dinesh Panday

Academic Editor

PLOS ONE